# Bacterial Cellulose Nanofibril-Based Pickering Emulsions: Recent Trends and Applications in the Food Industry

**DOI:** 10.3390/foods11244064

**Published:** 2022-12-15

**Authors:** Xingzhong Zhang, Dan Wang, Shilin Liu, Jie Tang

**Affiliations:** 1College of Food and Bioengineering, Xihua University, Chengdu 610039, China; 2College of Food Science & Technology, Huazhong Agricultural University, Wuhan 430070, China

**Keywords:** Pickering emulsion, bacterial cellulose nanofibrils, stabilization mechanism, nutrient encapsulation and delivery

## Abstract

The Pickering emulsion stabilized by food-grade colloidal particles has developed rapidly in recent decades and attracts extensive attention for potential applications in the food industry. Bacterial cellulose nanofibrils (BCNFs), as green and sustainable colloidal nanoparticles derived from bacterial cellulose, have various advantages for Pickering emulsion stabilization and applications due to their unique properties, such as good amphiphilicity, a nanoscale fibrous network, a high aspect ratio, low toxicity, excellent biocompatibility, and sustainability. This review provides a comprehensive overview of the recent advances in the Pickering emulsion stabilized by BCNF particles, including the classification, preparation method, and physicochemical properties of diverse BCNF-based particles as Pickering stabilizers, as well as surface modifications with other substances to improve their emulsifying performance and functionality. Additionally, this paper highlights the stabilization mechanisms and provides potential food applications of BCNF-based Pickering emulsions, such as nutrient encapsulation and delivery, edible coatings and films, fat substitutes, etc. Furthermore, the safety issues and future challenges for the development and food-related applications of BCNFs-based Pickering emulsions are also outlined. This work will provide new insights and more ideas on the development and application of nanofibril-based Pickering emulsions for researchers.

## 1. Introduction

Emulsions, as thermodynamically unstable systems containing two immiscible liquids of one liquid dispersed in another, have been widely applied in food fields containing beverages, dairy products, dressings, desserts, and sauces. They not only endow food products unique and various flavors, appearances, textures, and tastes, but also can be used as controlled delivery systems to encapsulate and protect nutrients or functional active ingredients [1,2,3]. The Pickering emulsion is a kind of emulsion first reported by Ramsden (1903) and Pickering (1907), formed by solid particles irreversibly adsorbed on the liquid–liquid or air–liquid interface [4,5]. Compared with conventional emulsions stabilized by low-molecular-weight surfactants and macromolecular emulsifiers, the surfactant-free Pickering emulsion has the unique potential advantages of high stability against droplet flocculation or coalescence, low toxicity and no irritation, high availability, and low cost [6]. 

Recently, food emulsions tended to develop in the direction of safety, sustainability, high stability, multifunctionality, and being green. Undoubtedly, Pickering emulsions stabilized by food-grade colloidal particles are more consistent with the relevant needs. Food-grade colloidal particles composed of various edible proteins, polysaccharides, lipids, waxes, and polyphenol crystals, are appropriate candidates as Pickering stabilizers, which have no adverse side effects with extensive sources and excellent biocompatibility [7]. With people’s increasing awareness of food security and nutrition, protein or polysaccharide food-grade particles have attracted more and more attention. Nanocellulose is a kind of eco-friendly cellulose material with at least one dimension in the nanoscale, consisting of cellulose nanocrystals (CNCs), cellulose nanofibers (CNFs), and bacterial cellulose (BC) [8,9]. Nanocellulose particles, with various aspect ratios and irregular shapes, have shown unique potential as stabilizers of Pickering emulsions, which have better emulsifying properties than the equivalent spherical particles and meet the requirements of being sustainable and environmentally friendly particle emulsifiers [10]. 

Bacterial cellulose (BC) is a natural ribbon-like nanofibril derived from microbial fermentation (Figure 1) and is classified as a “generally recognized as safe (GRAS)” dietary fiber by the USA Food and Drug Administration (FDA) [11]. As shown in Figure 2a, the individual nanofibrils in natural BC have an average diameter of 20–150 nm and several micrometer lengths, entangling and overlapping into stable network structures. BC has a high purity (>95%) without any pectin, lignin, or hemicellulose, which is convenient for separation and purification. Its crystallinity (>80%) is higher than that of plant-derived cellulose (40–78%), and it has a better hygroscopicity and water-holding capacity compared with conventional plant-derived cellulose [12]. It has various advantages, such as nontoxicity, non-allergenicity, low density, high surface area, good transparency, a nanoscale fibrous network, high mechanical strength and tensile strength, excellent cell adhesion, and proliferation (Figure 1), which are widely applied in food, cosmetics, pharmaceuticals, and so on [11,13]. It has the amphiphilicity to adsorb at the oil–water or air–water interfaces because of the many hydrophilic hydroxyl groups on the cellulose surface and the hydrophobic interactions from the crystalline organization, along with extensive hydrogen bonding of polymer chains [14,15]. Due to their excellent physical and chemical properties, BC-based nanoparticles are a promising clean-label candidate as colloidal particles to stabilize Pickering emulsions. 

In particular, cellulose nanofibrils take up almost 65% of the reported literature about nanocellulose and have been a key European bioeconomic priority since 2008 [19]. Interestingly, bacterial cellulose nanofibrils (BCNFs) as CNF particles have flexible nanosized fibrils with a large aspect ratio and can form strong entangled and disordered network structures, which perform unique advantages for stabilizing Pickering emulsions with high stability and good application prospects. Therefore, in this work, we focus on summarizing the fabrication, stabilization mechanism, features, and advanced applications of Pickering emulsions stabilized by BCNF-based nanoparticles. Finally, we state the potential challenges in food applications for BCNF-based Pickering emulsions and provide future outlooks on the research of nanofibril-based emulsions. 

## 2. Formation and Stabilization Mechanism of Pickering Emulsions

For Pickering emulsions, the particle’s surface wettability determines the particle’s adsorption site at the interface and the resultant emulsion’s type, which can be characterized by a three-phase contact angle (*θ*):(1)$$\cos\theta =\frac{\gamma_{PO}-\gamma_{PW}}{\gamma_{OW}}$$
where *γ_PO_*, *γ_PW_*, and *γ_OW_* are the surface tensions of the particle–oil, particle–water, and oil–water interfaces, respectively [20]. For hydrophilic particles with *θ* < 90°, including a majority of natural polysaccharides and protein particles, they tend to reside in the water phase and form the oil-in-water (O/W) Pickering emulsion. For hydrophobic particles with *θ* > 90°, they are close to the oil phase and stabilize the water-in-oil (W/O) Pickering emulsion. In addition, the fundamental difference between Pickering emulsions and conventional emulsions is that the Pickering particles attached to the interfaces can produce a strong detachment energy (Δ*E*). The Δ*E for* spherical colloidal particles can be calculated by the following equation [21]:(2)$$\Delta E_{sphere}=\gamma_{OW}\pi r^{2}{\left(1-\left|\cos\theta \right|\right)}^{2}$$
where *γ_OW_* is the oil–water interface tension (N/m), *r* is the particle radius (m), and *θ* is the contact angle of the particles. For rod-like particles, such as starch, chitin, and cellulose nanocrystals, the *ΔE* can be described as [22]: (3)$$\Delta E_{rod}=\gamma_{OW}lq{\left(1-\left|\cos\theta \right|\right)}^{2}$$
where *l* and *q* represent the length and width of the rod-like particles, respectively. When the adsorbed particles have a moderate wettability (*θ* approaches 90°) and a certain size (>10 nm), the Δ*E* is several orders of magnitude larger than the thermal energy (*k_B_*T) of Brownian motion, indicating that the particles are irreversibly adsorbed at the interfaces and result in a high stability of the stabilized Pickering emulsions.

In addition, the colloidal particles not only can stabilize the oil–water interfaces by reducing interface tension, but also can prevent droplet coalescence by causing steric hindrance and regulating the rheological properties [23]. For BC-based particles, they present different interfacial adsorption interactions and other stabilizations. The stability mechanism of Pickering emulsions stabilized by BC-based nanoparticles can also be summarized into the following four kinds: (1) the physical barrier effect of interfacial layers whereby a single layer or multiple layers formed by colloidal particles can effectively prevent droplet aggregation [24,25]; (2) a two- or three-dimensional (2D or 3D) network formation whereby the particle networks can hinder the contact and movement of droplets [26,27]; (3) depletion stabilization whereby a sufficiently high concentration of nonadsorbing polymers/particles can promote the flocculation of emulsion droplets and colloidal particles by generating osmotic stress [28,29]; and (4) capillary force interactions whereby the neighboring anisotropic particles create a strong capillary pressure to enhance the surface coverage and steric stabilization of the emulsion droplets [10].

## 3. Classification and Production of BC-Based Nanoparticles as Pickering Stabilizers

Recently, with the development of BC nanofabrication and functionalization, the preparation and application of Pickering emulsions stabilized by BC-based nanoparticles have attracted much attention. BC-based nanoparticles can be divided into bacterial cellulose nanocrystals (BCNCs) and nanofibrils (BCNFs) through mechanical treatment and/or acid hydrolysis [30]. These particles are produced from different approaches and have distinct sizes and shapes, emulsification, physical, and chemical properties, resulting in Pickering emulsions with varied characteristics. 

### 3.1. Bacterial Cellulose Nanocrystals (BCNCs)

BCNCs are rod-like nanoparticles produced from BC after selectively removing the amorphous region, which have a high crystallinity and a rigid structure with a length of 100–1000 nm and width of 10–50 nm [31]. Various approaches have been used to prepare BCNCs, including enzymatic hydrolysis, acid hydrolysis, ionic liquids, and oxidation treatment. Among these, acid hydrolysis is the most common method to prepare BCNCs, utilizing strong acids to selectively remove the amorphous components and split the crystalline microfibrils of BC, such as sulfuric acid (H_2_SO_4_), hydrochloric acid (HCl), and phosphoric acid (H_3_PO_4_) [32,33]. As shown in Figure 3a, the BC gelatinous membranes cultivated from *Acetobacter xylinum* were disintegrated into a uniform BC aqueous suspension by high-speed shearing and then were hydrolyzed by 50 wt% sulfuric acid at 45 °C for 3 h, followed by oxidation with 30 wt% hydrogen peroxide to obtain BCNC suspensions. In comparison to original BC, the olive oil Pickering emulsion stabilized by BCNCs presented a high thermal stability and a better emulsifying performance, while BCNC-stabilized emulsions were more sensitive to the change in pH and ionic strength [15]. In addition, Kalashnikova et al. produced BCNCs via acid hydrolysis with 2.5 M HCl at 70 °C for 2 h and applied them to stabilize an O/W Pickering emulsion. In Figure 2b, the obtained BCNCs had an average length of 855 nm and a width of 17 nm from the TEM image. The BCNCs could irreversibly adsorb on the hexadecane/water interface (Figure 3b), maintaining the long-term stability of the emulsion with droplets around 4 μm in diameter at a low BCNC concentration (5.2 mg BCNCs/mL hexadecane). Meanwhile, a droplet surface coverage of 60% was defined as the minimum requirement to stabilize Pickering emulsions [17]. Nevertheless, the acid hydrolysis of BC is a time-consuming and costly process and produces a large amount of acid waste, which is not a green, sustainable, and low-carbon method to prepare cellulose nanoparticles for Pickering stabilization.

### 3.2. Bacterial Cellulose Nanofibrils (BCNFs)

Compared with BCNCs, BCNFs are green and environmental nanoparticles used as Pickering stabilizers, which are usually obtained by subjecting BC suspensions via mechanical approaches without too much acid waste and pollution. BCNFs are long and flexible nanofibers, containing both crystalline and amorphous regions, which are organized with fibrillar elements of 10–50 nm widths and several micrometer lengths (Figure 2c). Similar to CNF preparation methods, BCNFs can be prepared by traditional mechanical treatments, such as homogenization, microfluidization, grinding, and aqueous counter collision, combined with enzymatic degradation, mild acid hydrolysis, or 2,2,6,6-tetramethylpiperidine-1-oxyl (TEMPO) oxidation to reduce energy consumption and improve particle dispersibility [16,36,37]. Different kinds of BCNF-based particles with various preparation methods have been successfully applied for Pickering emulsion stabilization, which are summarized in Table 1.

In general, high-pressure homogenization (HPH) is a common technique used to produce BCNFs in which the effects of the treatment conditions, including homogenized pressure and time, on the resultant BCNFs and stabilized Pickering emulsions have been systemically investigated [48,49]. In Figure 3c, Li et al. applied the HPH process to prepare BCNFs and investigated the influence of the BCNFs’ size, content, and environmental factors (temperature, ionic strength and pH) on the stabilized O/W Pickering emulsions. The size of the prepared BCNFs was gradually decreased with the increasing number of homogenized cycles (10–80 times), and all the BCNF suspensions were shear-thinning pseudoplastic fluids. The BCNFs that underwent 40 homogenized cycles had the best emulsifying ability, and the stabilized emulsions presented excellent stability to resist different environmental conditions [34]. Furthermore, Pickering emulsions stabilized by BCNFs with different oil-phase volumes (50–75%) were fabricated, and the effects of the oil–water ratios, storage time, and environmental factors on the surface coverage, coalescence rate, and stability of the resultant emulsions was investigated. The particle size and surface coverage of all the emulsions had no significant changes at a wide range of temperatures (4–50 °C) and ionic strengths (0–100 mM), and rheology analysis revealed the gel-like elastic behaviors of the emulsions. The results indicated that the obtained Pickering medium internal-phase emulsions (MIPEs) and high-internal-phase emulsions (HIPEs) had excellent physical stability against coalescence because of the formation of 3D networks by the BCNFs [38]. 

Moreover, surface modifications of BCNFs have been extensively researched and applied for Pickering emulsion stabilization in order to reduce energy consumption during mechanical treatment processing and to improve their functionality. TEMPO-mediated oxidation is a mild chemical modification process to allow the selective oxidation of C6-primary hydroxyl groups on the cellulose surface into C6-carboxyl groups, which can introduce charges on the BCNFs’ surface and improve particle uniformity [52]. As shown in Figure 2d, the TEMPO-oxidized bacterial cellulose nanofibrils (TOBC) had an average width of 38.8 ± 3.26 nm and a length of a few microns, which were dispersed uniformly and formed a loose network [18]. In Figure 3d, Jia et al. used a TEMPO/NaBr/NaClO system to prepare TOBC with different oxidation degrees and successfully utilized them to stabilize O/W Pickering emulsions. With the increasing amount of the NaClO oxidant (2–10 mmol/g), the contact angles and sizes of the obtained TOBC nanoparticles were gradually decreased, while the carboxyl group content was significantly increased (0.58–1.15 mmol/g). In view of particle wettability and prepared emulsion stability, 2-TOBC with 0.58 mmol/g carboxyl content is the optimum stabilizer for Pickering emulsions [35]. In addition, the rheological behaviors of TOBC with diverse oxidation degrees and their effects on the stability of the formed emulsions were further explored. The results show that the 2-TOBC emulsion had the strongest elastic structure and thereby exhibited the highest emulsion stability [39]. Wu et al. studied the influences of the TOBC surface charge density, pH, and ionic strength on the stability and rheology properties of O/W Pickering emulsions. The emulsion stabilized by TOBC with 1.16 mmol/g carboxyl content had a better stability and a higher viscosity, attributed to the greater number of individual nanofibrils and the formation of a denser network structure. Moreover, the viscosity and elasticity of the emulsions decreased with an increasing pH from 3.0 to 11.0 and became higher with an increased NaCl concentration (25–100 mM) due to the formation of a tighter nanofibril network caused by charge screening [40]. These studies confirm that the oxidation degrees and surface charge densities of TEMPO-oxidized BCNFs have important effects on the rheological properties and stability of manufactured O/W Pickering emulsions, being beneficial to the development of BCNF-based emulsion products with an appropriate fluidity. 

### 3.3. BCNF-Based Complex Nanoparticles

BCNFs can interact with other inorganic particles or biomass materials (polyphenols, polysaccharides, or proteins) to assemble into unique complex structures, which can further improve their emulsification and provide new functionalities [51,53]. Firstly, the organic–inorganic hybrid particles and co-stabilized Pickering interfacial layers can be designed by BCNFs and inorganic particles, such as silica (SiO_2_), titanium dioxide (TiO_2_), and clay nanoparticles. Li et al. prepared hybrid BC–TiO_2_ nanoparticles by simple physical adsorption and used them as stabilizers to construct a stable O/W Pickering emulsion. The hybrid particles adsorbed at the oil–water interfaces improve the mechanical strength and functionality of the interfacial layers, subsequently endowing emulsions with a high stability as well as remarkable photocatalytic activity [49]. In addition, Cho et al. fabricated a water-in-silicone oil (W/S) Pickering emulsion stabilized by in situ interfacial coacervation of attractive hectorite nanoplatelets (AHNPs) and BCNFs. The bilayered coacervate interface formed by AHNPs and BCNFs via electrostatic interactions significantly enhanced the interfacial modulus, and the obtained emulsion exhibited solid-like behaviors due to the interconnection of emulsion droplets through BCNF bridging [48]. Moreover, BCNFs combined with polyphenols or hydrophobic alkyl chains to self-assemble into fiber-filled layers at the O/W interface can remarkably improve the interfacial viscoelasticity, structural stability of emulsions, and other functional properties [50,51]. 

Notably, BCNFs can noncovalently and/or covalently interact with protein or polysaccharide macromolecules/particles to form unique complexes as Pickering stabilizers, which significantly improve emulsifying properties and enhance the environmental stability and functionality of the produced emulsions [54,55]. Soybean protein isolate (SPI), as a common plant-derived globular protein, is widely utilized to interact with BCNFs to synergistically construct oil–water interfacial layers, aiming to improve the emulsifying ability, structural properties, and other shortcomings of the single component. In Figure 4a, Liu et al. prepared bacterial cellulose nanofibers/soy protein isolate (BCN/SPI) composite colloidal particles using the antisolvent method and used them as stabilizers to develop high-internal-phase emulsions (HIPEs) with a 75% oil phase. The results showed that the BCN/SPI particles were formed mainly through hydrogen bonds confirmed by FTIR analysis, and the addition of BCNs significantly improved the emulsifying capacity of SPI as a stabilizer for HIPEs [41]. Meanwhile, the influences of pH on the conformational states, interfacial adsorption characteristics, and emulsifying properties of SPI/TOBC electrostatic complexes were investigated. As shown in Figure 4b, when the pH was in the range of 2–3, stable positively charged SPI/TOBC complexes were formed by electrostatic attraction and irreversibly adsorbed on the oil–water interfaces to form dense layers. At a pH of 4–5, much thicker interfacial adsorption layers were formed by the SPI aggregates, whose elastic behavior was strengthened in the presence of TOBC. At a pH of 7–9, electrostatic repulsion forces among the SPI and TOBC facilitated a greater amount of SPI to adsorb on the oil–water interfaces, and the TOBC network in the aqueous phase enhanced the emulsion stability [56]. Animal source proteins such as gelatin and whey protein were also researched to improve BCNF wettability and interface behaviors. In Figure 4c, Wu et al. investigated the improvement of O/W emulsion stability through regulating the electrostatic interaction between TOBC and gelatin protein (GLT). The synergism of TOBC and GLT increased the apparent viscosity and improved the viscoelastic properties of the obtained emulsions. At a pH of 4.7, the emulsions stabilized by TOBC/GLT electrostatic complexes at a mixing ratio of 1:2.5 had the highest viscosity and gel strength [44]. In addition, food-grade polysaccharides can also combine with BCNFs to enhance their dispersion stability, surface charge density, and rheological properties in favor of stabilizing Pickering emulsions [45,47]. For example, Zhang et al. designed bacterial cellulose/carboxymethyl chitosan (BC/CCS) complexes with tunable assembled behaviors via electrostatic interactions and investigated their stabilization mechanism for O/W emulsions. In Figure 4d, under alkaline conditions (pH 9.6), the BC/CCS complexes induced the long-range depletion–stabilization effect to stabilize the emulsions, and the depletion effect was increased with the increasing BC content, which prevented emulsion droplets from coalescing and aggregating [46]. In brief, the complex colloidal particles formed by the combination of BCNFs and protein or polysaccharide polymers can exhibit a moderate wettability and excellent interfacial adsorption properties, subsequently providing BCNF-based emulsions with an excellent physical stability and ideal functionalities, which have unique research and application prospects.

## 4. Applications of BCNF-Based Pickering Emulsions in the Food Industry

BCNF-based Pickering emulsions exhibited long-term stability, versatile functional properties, and other potential advantages over conventional emulsions. Consequently, more and more researchers are paying attention to the potential applications of food-grade Pickering emulsions stabilized by BCNF-based colloidal particles in the food industry, and an overview of the common applications of these emulsions is shown in Figure 5. 

### 4.1. Nutrient Encapsulation and Delivery

Many hydrophobic nutrients, such as essential oils, carotenoids, and polyphenols, have numerous important biological functions, such as antioxidation, antitumor, anti-inflammation, and antimicrobial activities, but their intrinsic poor stability, low bioavailability, and water insolubility seriously restrict their applications in food fields [60,61,62]. BCNF-based Pickering emulsions are a good delivery system to encapsulate and protect hydrophobic nutrients, functional factors, and probiotics, which can dramatically improve their encapsulation stability and bioavailability. Shen et al. prepared HIPEs stabilized by bacterial cellulose nanofiber/soy protein isolate (BCNs/SPI) complex particles and applied them to the delivery of curcumin. The curcumin-loaded HIPEs stabilized by BCN/SPI particles presented a better encapsulation efficiency and antioxidant activity than that of oil systems and surfactant-stabilized emulsions. In vitro digestion experiments showed that the bioaccessibility of curcumin encapsulated in BCN/SPI-stabilized HIPEs was significantly increased to 30.54%, meaning that stable HIPEs stabilized by BCN/SPI particles are a promising way to encapsulate and deliver lipid-soluble bioactive compounds [63]. 

Moreover, it has been reported to combine the spray drying technique with BCNF-based Pickering emulsions to produce oil microcapsules, which can encapsulate and protect bioactive ingredients against adverse environmental conditions and provide microcapsule powders with excellent water dispersibility and stability. In Figure 5a, edible oil powders were successfully fabricated by spray drying Pickering emulsions stabilized by heat-induced soy protein isolate (HSPI) and BCNF particles, and their physical properties, encapsulation efficiency, and digestion properties were explored. The moisture content, flowability, and encapsulation efficiency of the spray-dried oil powders were improved by tuning the BCNF content and the oil-to-water ratios of the emulsions. In addition, the presence of BCNFs partially inhibited lipid hydrolysis, and oil powders with 0.1 wt% BCNFs exhibited optimum anti-digestibility with the minimum released FFA values of 72.5% [57]. Furthermore, vitamin E (VE)-loaded O/W emulsions synergistically stabilized by gelatin and BCNFs were prepared and then were used to obtain stable VE-loaded microcapsules by spray drying. The VE microcapsules in the presence of BCNFs presented an improved microencapsulation performance by a reduction in the surface oil (unpublished paper). These microencapsulation systems combining the spray drying technique with BCNF-based Pickering emulsions are a facile way to encapsulate and deliver fat-soluble vitamins and other functional liquid oils, such as deep-sea fish oil, flaxseed oil, and grapeseed oil. 

### 4.2. Lipid Digestion Modulation

In general, lipid digestion is an interfacial process involving various lipase/colipase and bile salts adsorbed on the surface of oil droplets to promote their digestion and hydrolysis within the human gastrointestinal tract (GIT). In view of a healthy diet for humans, delaying lipid digestion and improving the bioavailability of oil-soluble nutrients are the main purposes of lipid digestion regulation, which can be regulated by oil–water interfacial properties, droplets size, and the stability of emulsion systems [64,65,66]. Particularly, BCNF-based Pickering emulsions have attracted more attention in modulating lipid digestion and absorption due to their unique advantages, including enzyme-unresponsive stabilizers, responsive and tunable interfaces, and long-term stability [67]. As shown in Figure 5b, the digestion properties and mechanism of medium-chain triglyceride (MCT) oil-in-water (O/W) Pickering emulsions stabilized by BCNF/SPI electrostatic complexes were studied. The emulsions in the presence of BCNFs displayed a smaller amount of free fatty acids (FFAs) released than that of pure SPI-stabilized emulsions after in vitro simulated digestion, with a minimum amount of FFAs released of 58.0% at a BCNF/SPI ratio of 1:5. It was attributed to the fact that the compact interfacial layers formed by the BCNF/SPI complexes partially restricted interfacial displacement by bile salts or lipases, and the bridging structure and interconnected networks formed by the BCNFs reduced the surface area available for binding with lipases and bile salts [42]. Therefore, Pickering emulsions co-stabilized by BCNF-based complex nanoparticles can be designed to delay the lipid digestion in food, subsequently increasing satiety and reducing appetite. 

### 4.3. Inhibiting Lipid Oxidation 

Lipid oxidation is an adverse phenomenon usually taking place in fatty food products, which not only induces the formation of offensive odors and toxic products, but also decreases their nutritional quality [68,69]. There are considerable interests in inhibiting lipid oxidation by using Pickering emulsion systems, depending on the construction of the dense interfacial layers as well as antioxidant utilization, to constrict oxidation reactions of lipid free radicals [70,71]. In Figure 5c, SPI/TOBC complexes were applied as emulsifiers to stabilize canola oil-in-water Pickering emulsions, and the emulsions’ oxidative stability was explored through an accelerated oxidation experiment at 40 °C. Both the peroxide values (POVs) and the thiobarbituric acid reactive substance (TBARS) products of the emulsions were significantly lower than that of pure canola oil during the accelerated oxidation process and gradually decreased when the SPI/TOBC complex content increased. Two main reasons can explain the results: (1) the formation of 3D interfacial networks by the SPI/TOBC complex particles scavenged free radicals and hindered the contact between the oil and oxygen; and (2) the addition of BCNFs increased the emulsions’ viscosity and viscoelasticity, thereby inhibiting the movement of oxidative free radicals [18]. Therefore, BCNF-based Pickering emulsions can be developed to enhance emulsion oxidative stability and control lipid oxidation in emulsified food. 

### 4.4. Edible Solid Foams 

Edible solid foams with porous structures and large surface areas have enormous potential in food applications, such as expanded snacks, encapsulation and delivery of active compounds, interfacial catalysis of food enzymes, and food packaging [72,73,74]. Pickering emulsions with 3D-network structures are a natural template to produce edible solid foam, especially for Pickering HIPEs. It was reported that edible solid foams were fabricated via Pickering emulsion templating (Figure 5d). O/W Pickering emulsions co-stabilized by SPI/TOBC complexes were prepared and then freeze-dried to obtain solid foams. The prepared solid foams presented porous structures and novel mechanical properties with a great compression performance and had a low cytotoxicity and excellent biocompatibility [43]. In addition, BCNF-based Pickering HIPEs can also be applied to prepare porous foams with tunable structures and variable functionalities by regulating the emulsion droplet size and interfacial structures, which have great advantages in the fabrication of food-grade solid foams. 

### 4.5. Edible Coatings or Films

Recently, Pickering emulsions stabilized by nanocellulose particles have been widely studied to fabricate edible coatings or films for food active packaging. Similar to conventional CNC or CNF materials, the presence of BCNFs in Pickering emulsions enhance the mechanical properties of the resultant films and improve the films’ functionality, containing moisture and oxygen barrier capabilities, antimicrobial and antioxidant activity, and transparency [75,76,77,78]. In Figure 5e, Li et al. prepared edible oleofilms by casting beeswax-in-water Pickering emulsions formed by the physical hybrid particles of BCNFs and carboxymethyl chitosan (CCS). The obtained oleofilms exhibited excellent mechanical properties and water vapor barrier properties and could be easily redispersed into water to recover back to the emulsion state without additional high energy mixing, which shows considerable potential for packaging applications [58]. Moreover, BCNF-based Pickering emulsions incorporated with other antibacterial components such as essential oils and chitosan will further endow the obtained coatings or films with an excellent antimicrobial performance, which possess great commercial application values in food preservation. 

### 4.6. Fat Substitutes

Bacterial cellulose is a kind of intrinsic water-insoluble dietary fiber synthesized by microorganisms, which can be directly used as a food ingredient to produce low-calorie and low-cholesterol products. Therefore, post-processed BCNFs are a promising dietary fiber to design reduced-fat food products [79,80,81]. Pickering emulsions stabilized by BCNF-based particles can also be utilized as fat substitutes to cut calories and increase the dietary fiber content of food products. As shown in Figure 5f, Xie et al. successfully prepared O/W Pickering emulsions stabilized by water-insoluble dietary fibers consisting of BCNFs and bamboo shoot water-insoluble dietary fiber (BSDF) and used them as fat substitutes to prepare low-calorie chewy biscuits. The structure and texture properties of the biscuit dough and baked biscuits with different fat replacement rates (0–90%) by BCNF/BSDF-stabilized Pickering emulsions were assessed. Obviously, the presence of these dietary fibers increased the biscuit springiness and decreased hardness, owing to the formation of irregular fiber networks which changed the gluten network structure in the reduced-fat doughs. In addition, the baked biscuit with a 35% fat replacement rate by emulsions exhibited the optimum appearance and texture and had a reduced-fat energy of 12.5 kJ/g with a total amount of dietary fibers of 1.05 mg/g. This work suggests that BCNF-based Pickering emulsions show a great application prospect as fat substitutes in the development of low-calorie baked foods [59]. Moreover, Zhang et al. prepared concentrated Pickering emulsions (>50 wt% oil) stabilized by SPI/TOBC complexes with a high stability, elastic gel-like rheological properties, and deformability, which also provides some guidance for applications as fat substitutes to replace trans and saturated fats, reduce calories, and supply dietary fibers in food [82]. 

## 5. Challenges and Future Perspectives

In conclusion, we have a relatively clear understanding of the stabilization mechanisms and potential applications of food-grade Pickering emulsions stabilized by BCNF-based particles as well as the production and characteristics of BCNFs and their complex nanoparticles for Pickering particles. However, several challenges need to be further solved to facilitate the food applications of BCNF-based Pickering emulsions:(1)There is an urgent requirement to deeply explore the toxicity and allergenicity of BCNF-based particles and their stabilized Pickering emulsions to ensure their safety in food applications. Consequently, the safety issues of BCNFs and their complex particles, as well as the influences of BCNF-based Pickering emulsions on gastrointestinal function and intestinal microflora can be further investigated by in vitro and in vivo studies; (2)Facile and low-cost preparation methods of BCNF-based Pickering stabilizers need to be provided and must be convenient for extensive commercial production and applications. Meanwhile, cost-effective modification methods of BCNF-based particles should be explored to improve their emulsification and functionality; (3)More potential applications of BCNF-based Pickering emulsions can be studied in the food industry, such as 3D-printed food, Pickering interfacial catalysis for enzymes, and probiotic encapsulation. 

Eventually, we believe this review will provide the latest progress and new insights into the field of nanocellulose-stabilized Pickering emulsions. 

## Figures and Tables

**Figure 1 foods-11-04064-f001:**
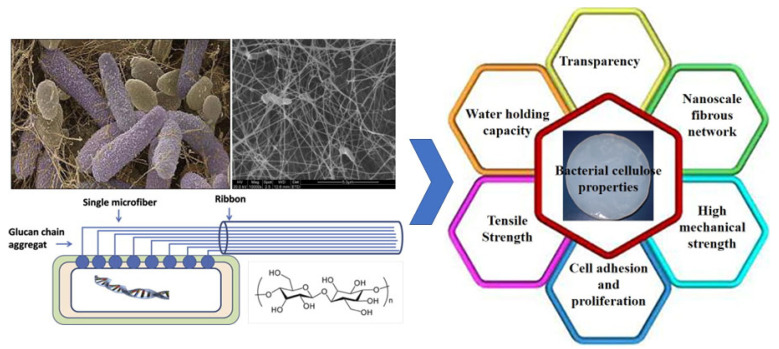
SEM image of bacterial cellulose formed by *Acetobacter xylinus* and its characteristics. Reprinted with permission from [11]. Copyright (2014), Elsevier.

**Figure 2 foods-11-04064-f002:**
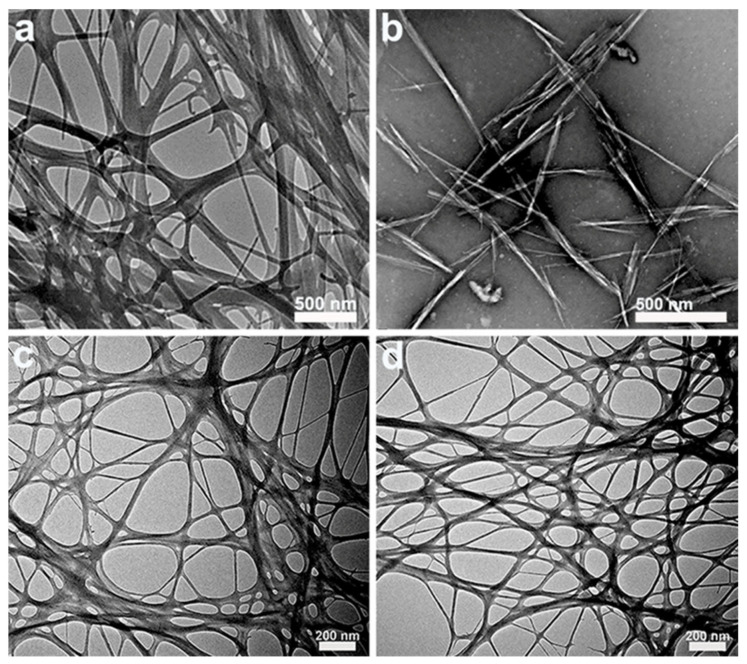
TEM images of (**a**) natural BC. Reprinted with permission from [16]. Copyright (2018), Elsevier. (**b**) BCNCs from BC by hydrochloric acid hydrolysis. Reprinted with permission from [17]. Copyright (2011), American Chemistry Society. (**c**) BCNFs from BC treated by high-pressure homogenization and (**d**) TOBC from BC treated by TEMPO-mediated oxidation. Reprinted with permission from [18]. Copyright (2019), Elsevier.

**Figure 3 foods-11-04064-f003:**
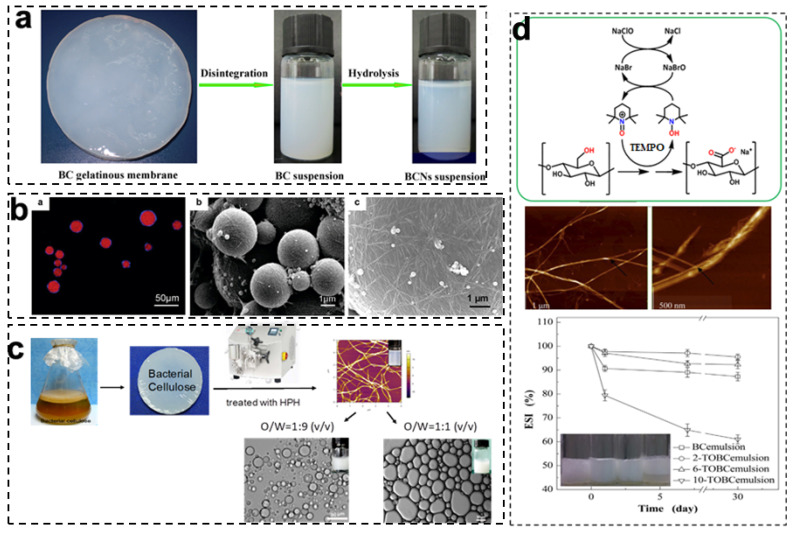
Studies on O/W Pickering emulsions stabilized by BC-based nanoparticles: (**a**) Pictures of the preparation process for BCNCs. Reprinted with permission from [15]. Copyright (2017), Elsevier. (**b**) CLSM images of emulsion droplets stabilized by BCNCs, and SEM images of a styrene Pickering emulsion stabilized by BCNCs. Reprinted with permission from [17]. Copyright (2011), American Chemistry Society. (**c**) Preparation process for BCNFs, and optical micrographs of a Pickering emulsion stabilized by BCNFs. Reprinted with permission from [34]. Copyright (2019), Elsevier. (**d**) Reaction pathway of TOBC produced by TEMPO oxidation, and the emulsion stability index (ESI) of TOBC-stabilized Pickering emulsions. Reprinted with permission from [35]. Copyright (2016), Elsevier.

**Figure 4 foods-11-04064-f004:**
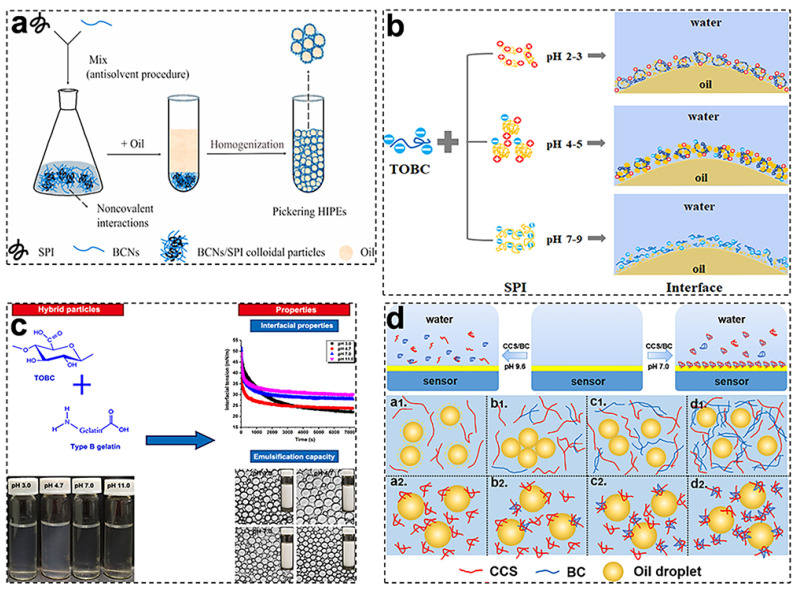
Studies on O/W Pickering emulsions stabilized by BCNF-based complex particles: (**a**) Schematic mechanism for the formation of Pickering high-internal-phase emulsions (HIPEs) stabilized by BCN/SPI composite colloidal particles. Reprinted with permission from [41]. Copyright (2021), Elsevier. (**b**) Schematic illustration of the interfacial adsorption behaviors of SPI/TOBC complexes at different pH values (2–9). Reprinted with permission from [56]. Copyright (2021), Elsevier. (**c**) Appearance and interfacial properties of TOBC–gelatin (GLT) mixtures, and morphology of TOBC/GLT-mixture-stabilized emulsions. Reprinted with permission from [44]. Copyright (2022), Elsevier. (**d**) Schematic illustration of the interfacial adsorption behaviors of BC/carboxymethyl chitosan (CCS) dispersions at the O/W interface investigated by QCM-D, and stabilization mechanism of BC/CCS-based emulsions: a, CCS only; b, CCS with a few BCs; c, CCS with many BCs; and d, CCS with very many BCs. The pH values were controlled between 9.6 (**a1**–**d1**) and 7.0 (**a2**–**d2**). Reprinted with permission from [46]. Copyright (2022), Elsevier.

**Figure 5 foods-11-04064-f005:**
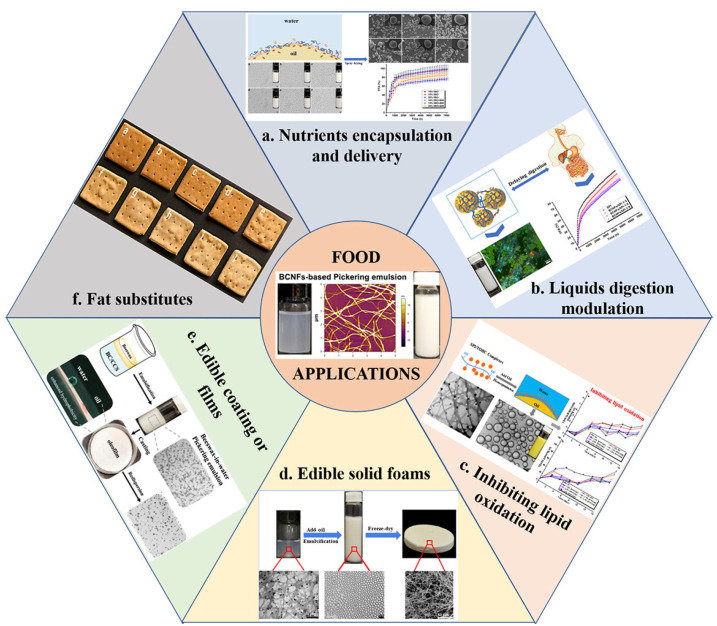
Overview of the applications of O/W Pickering emulsions stabilized by BCNF-based particles: (**a**) Nutrient encapsulation and delivery. Reprinted with permission from [57]. Copyright (2022), Elsevier. (**b**) Lipid digestion modulation. Reprinted with permission from [42]. Copyright (2022), Elsevier. (**c**) Inhibiting lipid oxidation. Reprinted with permission from [18]. Copyright (2019), Elsevier. (**d**) Edible solid foams. Reprinted with permission from [43]. Copyright (2020), Elsevier. (**e**) Edible coatings or films. Reprinted with permission from [58]. Copyright (2020), American Chemistry Society. (**f**) Fat substitutes. Reprinted with permission from [59]. Copyright (2021), Elsevier.

**Table 1 foods-11-04064-t001:** Summary of the preparation of Pickering emulsions using different BCNF-based colloidal particles.

Particle Type		Preparation Method	Size (d = Diameter, h = Height, w = Width, I = Length)	Particle Content	Emulsification Method	Emulsion Type	Oil-Phase Type and Content	References
BCNFs		High-pressure homogenization	6 < h < 13 nm 97 < w < 127 nm I < 10 μm	0.1–0.5 wt%	High-speed shear, 10,000 rpm, 2 min	O/W	Dodecane (10–50 v%)	[34]
		High-pressure homogenization	3.8 < h < 8.9 nm 66.8 < w < 128.6 nm	0.1–0.5 wt%	High-speed shear, 12,000 rpm, 2 min	O/W	Dodecane (50–75 v%)	[38]
		Hydrochloric acid hydrolysis	30 < d < 80 nm	0.05% (*w*/*v*)	High-speed shear, 15,000 rpm, 1 min, and high-pressure homogenization, 600 bar, 1 min	O/W	Peanut oil (5–30 v%)	[16]
		TEMPO oxidation	5 < w < 50 nm I < 5 μm	0.18–0.70 wt%	Ultrasound emulsification, 40 W, 5 min, 50% ultrasonic pulse	O/W	Liquid paraffin (50 v%)	[35,39]
		High-pressure homogenization and TEMPO oxidation	5.7 < h < 10.8 nm 70.3 < w < 101.6 nm	0.3 wt%	Ultrasound emulsification, 20 kHz, 95% amplitude, 2 min	O/W	Dodecane (10 v%)	[40]
BCNF-based complexes								
BCNF-protein	BCNF-soy protein isolate (SPI)	Antisolvent precipitation approach	927.0 < d < 1510.3 nm	2.0 wt%	High-speed shear, 20,000 rpm, 3 min	O/W	Sunflower seed oil (75 v%)	[41]
	BCNF-SPI	Electrostatic interaction	10.3 < h < 19.1 nm 130.9 < w < 160.7 nm I < 10 μm	1.0 wt%	High-speed shear, 16,000 rpm, 2 min	O/W	Medium-chain triglyceride (MCT) oil	[42]
	TEMPO-oxidized BCNF (TOBC)-SPI	Electrostatic interaction	8.7 < h < 14.9 nm 102.4 < w < 121.6 nm I < 10 μm	1.08-3.24 wt%	High-speed shear, 16,000 rpm, 2 min	O/W	Dodecane (50–74.05 v%)	[18,43]
	BCNF-gelatin (GLT)	Electrostatic interaction		1.0 wt%	High-speed shear, 10,000 rpm, 2 min	O/W	Dodecane (75 v%)	[44]
BCNF-polysaccharide	BCNF-sodium alginate (SA)	High-pressure homogenization and mixing	6.8 < h < 7.6 nm 55.0 < w < 60.2 nm	0.1–0.5 wt%	Ultrasound emulsification, 20 kHz, 3 min	O/W	Dodecane (10–30 v%)	[45]
	BCNF-carboxymethyl chitosan (CCS)	Electrostatic interaction and depletion effect		0.01-0.20 wt%	High-speed shear, 22,000 rpm, 4 min	O/W	Soybean oil (50 v%)	[46]
	BCNF-carboxymethyl cellulose (CMC)	Mixing and spray drying	581.3 < d < 620.7 nm	0.1–0.5 wt%	High-speed shear, 20,000 rpm, 4 min	O/W	Isohexadecane (10 v%)	[47]
BCNF-attractive hectorite nanoplatelets (AHNPs)		High-pressure homogenization and TEMPO oxidation	I ≈ 3 μm	0.65 wt%	High-speed shear, 15,000 rpm for 10 min	O/W & W/O	silicone oil (10–90 v%)	[48]
BCNF-titanium oxide (TiO_2_)		High-pressure homogenization and ultrasonic treatment		0.2–0.6 wt%	High-speed shear, 20,000 rpm, 2 min	O/W	Dodecane (10–80 v%)	[49]
BCNF-octadecylamine (C18)		TEMPO oxidation and grafting	5 < w < 10 nm	0.05–0.1 wt%	High-speed shear, 15,000 rpm, 10 min	O/W	N-decane; olive oil; fomblin oil; silicone oil (10–40 v%)	[50]
BCNF-tea polyphenols (TPs)		High-pressure homogenization and physical adsorption		0.1–0.5 wt%	High-speed shear, 10,000 rpm, 2 min	O/W	Dodecane; camellia seed oil (10 v%)	[51]

## Data Availability

The datasets generated for this study are available on request to the corresponding author.

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
