# Peer review of "Bacterial Cellulose Nanofibril-Based Pickering Emulsions: Recent Trends and Applications in the Food Industry"

_foods, 2022, doi:10.3390/foods11244064_

Round 1

Reviewer 1 Report

Review of manuscript "Bacterial cellulose nanofibrils based Pickering emulsions: Recent trends and applications in food industry" by Xingzhong Zhang, Dan Wang, Shilin Liu and Jie Tang

Abstract reflects the content of review manuscript in appropriate manner.

Keywords: keyword ”application” – does not have a sense. I would suggest add more specific food-industry related keywords, otherwise almost all the given keywords can be found in the title of review. Adding different keywords would improve the searchability.

Review has been divided in 5 sections, the structure of review is justified and logic.

Introduction

Definition of emulsion system in general and Pickering emulsion specifically, as well as their application examples for food industry have been given in the beginning of the manuscript. Development of green, safe, sustainable Pickering emulsions containing food-grade protein of polysaccharide colloidal particles has been set as one of the main tendencies in food emulsions area because of their relevance to the needs of society. Nanocellulose has been highlighted as appropriate sustainable and environmentally friendly particle emulsifiers. Description of bacterial cellulose has been given as well as its properties and applications, concluding with statement of BC-nanoparticle as a promising candidate for Pickering emulsion. Good justification of the review manuscript is given in the end of introduction, giving clear vision of the reason, structure and content of the presented review.

Theory of formation and stabilization mechanism of Pickering emulsion has been comprehensively explained in section 2.

Classification of BC nanoparticles is based on the preparation method and resulting properties of particles respectively. Furthermore, these parameters have significant impact on the properties of Pickering emulsions.

BNCFs are nominated to be more appropriate candidates for Pickering emulsions because of greener and more sustainable preparation methods without involving of acids.

Comprehensive Table (Table 1) has been given to summarize articles on the topic.

Some individual articles on BNCFs in Pickering emulsions have been explored more in order to introduce and to explain most significant BNCFs properties, important for emulsion properties.

Food application topic have been comprehensive reviewed, sub-topics are divided into sub-sections.

Section 5 is great benefit of review, summarizing challenges and future perspectives of BCNF application in food industry.

The review contains figures with representative illustrations of described researches and processes, it is a benefit.

Comments:

Only two articles have been cited regarding BCNC particles in Pickering emulsions. Nevertheless, the review focuses on BCNF, it should be explained, if only two researches exist or it is a choice of authors to cite just few.

Line 82/ Line 152 and in other places in manuscript: Phrases “we investigated” and “we have combined” are somehow inappropriate for the review article, even if the authors refer to their own work.  It is better to use neutral review of articles and researches without highlighting your own authorship.

Author Response

Point 1: Keywords: keyword “application” - does not have a sense. I would suggest add more specific food-industry related keywords, otherwise almost all the given keywords can be found in the title of review. Adding different keywords would improve the searchability.

Response 1: Good suggestion! We have revised it.

Comments:

Point 2: Only two articles have been cited regarding BCNC particles in Pickering emulsions. Nevertheless, the review focuses on BCNF, it should be explained, if only two researches exist or it is a choice of authors to cite just few.

Response 2: Thanks for your valuable questions. This review mainly introduced the BCNFs-based Pickering emulsions, so we cited few articles about BCNCs particles for Pickering emulsions stabilization.

Point 3: Line 82/ Line 152 and in other places in manuscript: Phrases “we investigated” and “we have combined” are somehow inappropriate for the review article, even if the authors refer to their own work.  It is better to use neutral review of articles and researches without highlighting your own authorship.

Response 3: Thanks for your good suggestions. We have revised these statements, and make neutral review of articles and researches over the paper.

Reviewer 2 Report

General comments:

This document presents a review on the use of bacterial cellulose (BC), in its various forms (through physical and chemical modification, including surface modifications) in Pickering emulsions. While in the abstract (lines 15 and 18) and introduction (line 84) the authors aimed at providing information on the physicochemical properties and stabilization mechanisms of BC, there is little to none information in these topics. Section 2 presents only a general summary on the formation and stabilization mechanism, not going into more details in the reviewed literature. More information on the above would improve the quality of the document. Also, a full review of the document to correct for small but systematic grammar errors is advised.

Specific comments:

Figure 1 Species names in italic

Figure 1. I don’t see the advantage of the schematics of the BC properties, since they are already mentioned in the text (Page 2, line 63).

Also, I don’t see the advantages of figures 2c and 2d as no comments are made in the introduction text regarding the treatments. Figure 2d shows TOBC treated fibres but the meaning of TOBC only shows up in page 8, line 32.

As a suggestion, perhaps figure 1 (left side) coud ne merged with figures 2a (eventually 2b as well), in a single figure. Alternatively, figures 2b,c,d could be added to figure 3

3.2. I agree that the mechanical processes are simpler and the use of lower amounts and/or concentration of chemicals are more environmentally desirable. I understand the arguments of sustainability and cost, but, as a suggestion, I think it is wiser to stay away from such concepts, unless  some references are (at least) provided on environmental sustainability (e.g. life cycle assessment) and economics. In the document there is not enough information to better support such claims (which to some extent are understandable, since it is a review paper on the stabilization of emulsions with BC).

Page 8, lines 1-2. A few references to support the sentence “…high-pressure homogenization (HPH) is a common technique…” could be added. Only two references are mentioned (35, 36) while they are related to the preparation of emulsions.

Fig 4 is of poor quality (low resolution) in the pdf document. The same in figure 5. In both, small lettering is difficult/impossible to read.

Page 11, line 152 “Moreover, we have combined spray drying technique with….” Concerns with ref 59. The authors of that reference are not the same of the submitted manuscript. The same for line 181 (“we investigated”) in page 12, relating to ref 64. Is “we” the correct pronoun?. The same in section 4.3, line 199, section 4.4, line 216 and perhaps more. The authors should review the document and make the necessary amendments.

Author Response

Point 1: This document presents a review on the use of bacterial cellulose (BC), in its various forms (through physical and chemical modification, including surface modifications) in Pickering emulsions. While in the abstract (lines 15 and 18) and introduction (line 84) the authors aimed at providing information on the physicochemical properties and stabilization mechanisms of BC, there is little to none information in these topics. Section 2 presents only a general summary on the formation and stabilization mechanism, not going into more details in the reviewed literature. More information on the above would improve the quality of the document. Also, a full review of the document to correct for small but systematic grammar errors is advised.

Response 1: Thank you for your valuable comments. Firstly, we presented a general summary of formation and stabilization mechanism of BCNFs-based Pickering emulsions. Certainly, the detailed stabilization mechanisms of Pickering emulsions stabilized by BCNFs-based particles had been discussed in the section 3, such as 3D network (Page 8, line 18-19); high viscosity and denser network structure (Page 8, line 41-46); bilayered coacervate interface and BCNFs bridging (Page 8, line 64-66); moderate wettability and excellent interfacial adsorption (Page 9, line 107-108). Moreover, we improved the quality of the manuscript carefully, and response to the reviewers’ comments in the following section (point by point).

Point 2: Figure 1 Species names in italic

Response 2: We have revised it.

Point 3: Figure 1. I don’t see the advantage of the schematics of the BC properties, since they are already mentioned in the text (Page 2, line 63).

Response 3: We just summarized the BC advantages in the Figure 1, and presented the advantages of BC in the introduction.

Point 4: Also, I don’t see the advantages of figures 2c and 2d as no comments are made in the introduction text regarding the treatments. Figure 2d shows TOBC treated fibres but the meaning of TOBC only shows up in page 8, line 32.

Response 4: We have introduced the advantages of BCNFs and TOBC in the introduction (Page 3, line 80-89), and presented their treatments in section 3 (Page 5, line 170-178). The TOBC particles is a kind of BCNFs-based fibers produced by TEMPO oxidation, so we classified and introduced it in the section 3 of BCNFs-based particles.

Point 5: As a suggestion, perhaps figure 1 (left side) could be merged with figures 2a (eventually 2b as well), in a single figure. Alternatively, figures 2b,c,d could be added to figure 3

Response 5: Thanks for your good suggestions. But I think that the schematics of Figures 1 and 2 are appropriate and have a moderate size, so they are not need to be merged.

Point 6: 3.2. I agree that the mechanical processes are simpler and the use of lower amounts and/or concentration of chemicals are more environmentally desirable. I understand the arguments of sustainability and cost, but, as a suggestion, I think it is wiser to stay away from such concepts, unless  some references are (at least) provided on environmental sustainability (e.g. life cycle assessment) and economics. In the document there is not enough information to better support such claims (which to some extent are understandable, since it is a review paper on the stabilization of emulsions with BC).

Response 6: Thanks for your good suggestions, we have revised it.

Point 7: Page 8, lines 1-2. A few references to support the sentence “…high-pressure homogenization (HPH) is a common technique…” could be added. Only two references are mentioned (35, 36) while they are related to the preparation of emulsions.

Response 7: Good suggestion. We have supplied it, and the relevant references to support “…HPH is a common technique…” are summarized in Table 1 (Preparation method).

Point 8: Fig 4 is of poor quality (low resolution) in the pdf document. The same in figure 5. In both, small lettering is difficult/impossible to read.

Response 8: Good suggestion. We have revised Figure 4 and Figure 5.

Point 9: Page 11, line 152 “Moreover, we have combined spray drying technique with….” Concerns with ref 59. The authors of that reference are not the same of the submitted manuscript. The same for line 181 (“we investigated”) in page 12, relating to ref 64. Is “we” the correct pronoun?. The same in section 4.3, line 199, section 4.4, line 216 and perhaps more. The authors should review the document and make the necessary amendments.

Response 9: Thank you for your valuable questions. We have revised it.
